# A Multifunctional Dental Resin Composite with Sr-N-Doped TiO_2_ and n-HA Fillers for Antibacterial and Mineralization Effects

**DOI:** 10.3390/ijms24021274

**Published:** 2023-01-09

**Authors:** Yuanhang Zhao, Hong Zhang, Lihua Hong, Xinying Zou, Jiazhuo Song, Rong Han, Jiawen Chen, Yiyan Yu, Xin Liu, Hong Zhao, Zhimin Zhang

**Affiliations:** 1Department of Endodontics, School of Dentistry, Jilin University, Changchun 130021, China; 2Jilin Provincial Key Laboratory of Tooth Development and Bone Remodeling, Changchun 130021, China

**Keywords:** dental resin composite, dental caries, secondary caries, titanium dioxide, antibacterial property, remineralization

## Abstract

Dental caries, particularly secondary caries, which is the main contributor to dental repair failure, has been the subject of extensive research due to its biofilm-mediated, sugar-driven, multifactorial, and dynamic characteristics. The clinical utility of restorations is improved by cleaning bacteria nearby and remineralizing marginal crevices. In this study, a novel multifunctional dental resin composite (DRC) composed of Sr-N-co-doped titanium dioxide (Sr-N-TiO_2_) nanoparticles and nano-hydroxyapatite (n-HA) reinforcing fillers with improved antibacterial and mineralization properties is proposed. The experimental results showed that the anatase-phase Sr-N-TiO_2_ nanoparticles were synthesized successfully. After this, the curing depth (CD) of the DRC was measured from 4.36 ± 0.18 mm to 5.10 ± 0.19 mm, which met the clinical treatment needs. The maximum antibacterial rate against *Streptococcus mutans* (*S. mutans*) was 98.96%, showing significant inhibition effects (*p* < 0.0001), which was experimentally verified to be derived from reactive oxygen species (ROS). Meanwhile, the resin exhibited excellent self-remineralization behavior in an SBF solution, and the molar ratio of Ca/P was close to that of HA. Moreover, the relative growth rate (RGR) of mouse fibroblast L929 indicated a high biocompatibility, with the cytotoxicity level being 0 or I. Therefore, our research provides a suitable approach for improving the antibacterial and mineralization properties of DRCs.

## 1. Introduction

Dental caries, which is the most prevalent worldwide disease, is ranked as the third largest non-infectious disease by the World Health Organization [1]. In the past several decades, dental caries has received substantial interest in numerous studies and initiatives. Among these research studies, dental resin composite (DRC) is an effective method in repairing decayed teeth because of its excellent esthetics and convenient operation qualities [2,3,4], which enable more precise polishing and exhibit better wear resistance. Note, however, that in long-term dental resin restoration, the growing bacteria will create oral biofilms on the resin’s surface, particularly the gap formatted by polymerization shrinkage, and continuously degrade the current resin composite and dental tissue nearby, causing secondary caries and treatment failure [5,6]. To satisfy current clinical requirements and lengthen the longevity of tooth restorations, innovative and multifunctional dental resin composites are urgently needed.

DRC is mainly composed of a resin matrix and fillers; thus many researchers have concentrated on the development of these components [7,8]. For instance, Lucas et al. designed a dental adhesive consisting of polyhexamethylene guanidine hydrochloride (PHMGH) with antibacterial activity [9]. Zhou et al. developed a novel antibacterial resin containing a low concentration of dimethylaminohexadecyl methacrylate (DMAHDM), which could inhibit biofilms from saliva [10]. Unfortunately, the physicochemical qualities may be impaired due to the long alkyl chain of quaternary ammonium compounds based on the above approaches of resin matrices [11]. For the filler section, the functional fillers can greatly enhance the performances or impose new functions on DRC [12], such as the strong antibacterial effects [13,14] of ZnO nanoparticles [15]. It is worth noting that metallic and oxide particles can change the shade of materials, influencing the color and light transmission of the DRC accordingly [16,17]. The cumulative toxic effects of the long-term released ions also deserve further investigation [18,19].

In particular, as a photocatalytic antibacterial agent filler, titanium dioxide (TiO_2_) has attracted much attention in the field of dental materials due to its excellent performance [20]. TiO_2_ can be activated by absorbing ultra-high bandgap energy radiation [21]. Then reactive oxygen species (ROS) are generated, which have an antibacterial impact on bacteria as a result of the holes formed in the activated TiO_2_ [22]. Based on this superior efficacy of TiO_2_, Akira et al. proposed a resin composite with a TiO_2_ filler, which demonstrated enhanced antibacterial properties [23]. However, the wide bandgap of TiO_2_ requires UV radiation [24,25], which restricts its potential for additional clinical use. To address this weakness, several approaches, including doping with other species, have been thoroughly investigated to inspire TiO_2_ under visible light irradiation. For example, TiO_2_ has been doped with nitrogen (N) in earlier studies [26,27]. When exposed to visible light, N-doped TiO_2_ displays a narrowing bandgap and rising absorption, and more ROS are released concurrently to increase the antibacterial activity [28]. In addition, Costa et al. increased the photocatalytic activity by substituting strontium (Sr) for Ti in TiO_2_ [29], which can be explained in terms of the lattice deformation brought about by the higher radius of Sr [30]. More recently, with further research studies on modified TiO_2_, there has been a lot more interest in the co-doping of TiO_2_, which may produce more significant photocatalytic activity compared to single-element doping. According to Zhao’s study, co-doping copper (Cu) and N in TiO_2_ caused a narrower bandgap and more visible region of light absorption compared to TiO_2_ which was only doped with N [31]. This has rarely been reported for the application of Sr-N-co-doped TiO_2_ (Sr-N-TiO_2_) nanoparticles in dentistry.

However, the current modified methods of DRC have been mostly focused on a single function, which is far from optimal. Thus, there remains a need for more multifunctional DRCs. Hydroxyapatite (HA) has been noted as a suitable inorganic filler for bionic dental restoration materials. HA is a critical constituent of bone and dental tissues in the human body [32], which can release calcium and phosphorus ions to promote remineralization and repair the demineralized areas [33,34]. Par et al. proved that DRC containing HA could form a precipitate layer on the surface after being immersed in artificial saliva (SBF) [35]. Meanwhile, HA–TiO_2_ composite was synthesized, which was a new type of photocatalyst and exhibited excellent degradation performance for decontamination [36,37]. These above studies motivated us to fuse TiO_2_ and HA to synthesize a multifunctional DRC. 

In this study, a novel and multifunctional dental resin composite composed of Sr-N-TiO_2_ and HA reinforcing fillers was proposed. Specifically, the anatase-phase Sr-N-TiO_2_ was synthesized by a sol-hydrothermal process. With the help of these distinctive elements, the TiO_2_ was capable of activation under visible light irradiation and improved its antibacterial properties. To demonstrate the resin’s performance, the resin was thoroughly characterized in terms of its antibacterial activity, bioactive mineralization capacity, biocompatibility, and the physiochemical properties necessary for its application in restoration. Based on the research and analysis of the experimental results, the proposed DRC offers a fresh approach for clinical restoration and has potential in practical clinical applications. By simultaneously eliminating cariogenic bacterial species and remineralizing new minerals at the restoration margin, the combined antibacterial and mineralization therapy seemed to be more ideal.

## 2. Results

### 2.1. Characterization of Sr-N-TiO_2_ Powders

Figure 1A,B shows the nanoparticle morphology of the powder by field emission scanning electron microscopy (FE-SEM) and transmission electron microscopy (TEM). The particles are slightly agglomerated. Lattice spacings of 0.309 nm were visible in the high-resolution TEM (HRTEM) image (Figure 1C), which corresponded to the (101) facets of the anatase phase of TiO_2_. The energy-dispersive X-ray spectroscopy (EDS) maps in Figure 1D,E demonstrate a uniform distribution and show the proportion of each element in the nanoparticle.

To confirm the composition of the treated samples and to determine the electronic states and bonding configuration, X-ray photoelectron spectroscopy (XPS) measurements were conducted. Both Ti and O were detected in the doped TiO_2_ samples (Figure 1F,G). The three prominent and characteristic peaks were observed. The first peak at 529.7 eV was attributable to O 1 s. Another two peaks located at 458.5 eV and 464.2 eV were allocated as Ti 2p3/2 and Ti 2p1/2. Moreover, three additional peaks were observed. Two peaks at 133.1 eV and 134.7 eV could be ascribed to the Sr 3d3 and Sr 3d5, and the weak peak at 399.6 eV was designated as N 1 s. Obviously, the XPS results strongly proved that the Sr and N elements were doped into TiO_2_.

The prominent characteristic peaks of the anatase phase structure of TiO_2_ were visible in the X-ray diffraction (XRD) patterns (Figure 1H). The planes (101) were recognized as the distinct diffraction peaks of the anatase phase of TiO_2_ and were all in good agreement with PDF#78-2486. It showed that the samples were mainly single-phase structures. The SrO phase diffraction peaks were not observed in the XRD patterns, indicating that Sr exists in the form of holocrystalline substitution or an amorphous state in the material. Most half-width peaks of the doped TiO_2_ were significantly more significant than those of the undoped TiO_2_. The Scherrer formula revealed that the lattice parameters of the doped TiO_2_ samples were 9.47 nm, which was marginally less than the 11.13 nm of the undoped TiO_2_. This also indirectly indicates that Sr and N were doped into TiO_2_. 

Figure 1I illustrates that Raman active bands are shifted to 146, 399, 518, and 642 cm^−1^, due to the new linkages by the incorporated elements. The bars respectively correspond to the E_g_, B_1g_, A_1g_, B_1g_, and E_g_ modes of the anatase phase of TiO_2_. The Raman bands were well in accordance with the XRD profile.

We measured the UV–Vis absorption spectrum to look into the photoelectrochemical performance. When compared to pure TiO_2_, the absorption margins of the doped samples were extended into the visible range and shifted to longer wavelength regions, as seen in Figure 1J.

### 2.2. Properties of DRCs

The DRCs were synthesized. The formulations of fillers in each group are shown in Table 1. The reinforcing fillers were composed of an equal ratio of Sr-N-TiO_2_ and n-HA. The traditional fillers were SiO_2_. The DRCs were obtained by photopolymerization under the standard conditions for curing dental materials. They were examined for subsequent properties.

#### 2.2.1. Physicochemical Properties

The FT-IR results were plotted in Figure 2A, and the two feature peaks were marked. Figure 2B–D and Table 2 showed the degree of conversion (DC) in each group at 20 s, 40 s, and 60 s. With the same curing time, the DC rate of each group decreased with the increase in the filler ratio. With the proliferation in curing time, the DC rate of all groups showed an increasing trend. The statistical results were revealed. When curing for 20 s, compared with 47.12 ± 0.56% of the 0% group, the DC rates of the 2.5% group, 5% group, and 7.5% group were 40.84 ± 2.30% (*p* < 0.01), 37.99 ± 1.50% (*p* < 0.001), and 34.75 ± 1.44% (*p* < 0.0001), respectively. When curing for 40 s, compared with 50.13 ± 2. 06% of the 0% group, the DC rates of the other three groups were 46.71 ± 2.93% (*p* > 0.05), 44.39 ± 0.37% (*p* < 0.05), and 43.21 ± 2.13% (*p* < 0.05). After curing for 60 s, the differences between the groups gradually narrowed. Compared with 55.04 ± 1.21% of the 0% group, the DC rates of the other three groups were 54.30 ± 2.83% (*p* > 0.05), 52.27 ± 3.80% (*p* > 0.05), and 48.98 ± 1.13% (*p* < 0.05).

According to the standard, the curing depth (CD) of DRCs was calculated (Figure 2E and Table 2). The CD also decreased with the increase in the filler ratio, in line with the trend of the DC rate. The CD of the experimental resins ranged from 4.36 ± 0.18 mm to 5.10 ± 0.19 mm. 

The water contact angle (WCA) results are shown in Figure 2F,G. The WCA values range from 87.82 ± 3.22° to 92.72 ± 4.14°. These differences were statistically insignificant, except that the WCA values of the 0% group were statistically significantly lower than that of the 5% group (*p* < 0.01).

#### 2.2.2. Antibacterial Properties and Antibacterial Mechanism

The facts of the colonies on the bacterial adhesion plate are displayed in Figure 3A,B. The antibacterial rates (AR) in each group were 0% (448.33 ± 23.71), 47.43% (235.67 ± 11.50), 94.13% (26.33 ± 4.04), and 98.96% (4.67 ± 2.08). Compared with the 0% control group, the number of colonies was significantly lower in the other three groups (*p* < 0.0001). In the experimental groups, the number of colonies was significantly reduced in the 5% and 7.5% groups compared to the 2.5% group (*p* < 0.0001); there was no significant difference between the 5% and 7.5% groups. 

The biofilm metabolic activity was demonstrated by crystal violet staining (Figure 3C). The relative optical density (OD) values of the experimental groups were 33.83%, 26.59%, and 16.64% (*p* < 0.0001). So, the corresponding relative clearance rates were calculated to be 66.17%, 73.41%, and 83.36%.

Figure 3D indicates the changes in the absorbance values in the diphenylisobenzofuran (DPBF) reaction solution. The absorption peak of DPBF at 410 nm was progressively reduced with the curing time.

In Figure 3E,F, the colony amounts of each group were 629 ± 19.47, 566.67 ± 7.77, 92.67 ± 24.50, and 425.67 ± 8.08. The colony counting in positive group with Sr-N-TiO_2_ had statistical difference compared with that of the other three groups (*p* < 0.0001). Compared with the blank group and the negative group, the CFU counting of the experimental group was significantly decreased (*p* < 0.0001).

The FE-SEM images show the bacterial morphology, exhibiting different degrees of damage to the cell membrane structure (Figure 3G). In the control group, a large number of bacteria were observed, and the cell membrane was complete and smooth. In the experimental groups, the number of bacteria gradually decreased, and the bacterial morphology was wrecked at various levels. In the 2.5% group, the membrane turned rough, irregular, and somewhat concave. Cytoplasmic leakage, bacterial disintegration, and severe cell wall destruction appeared in the 5% and 7.5% groups.

The live/dead fluorescence test of *S. mutans* attached to the resin surface is depicted in Figure 4. During imaging, the damaged bacteria were stained red, the living bacteria were stained green, while the overlapping areas of live and dead bacteria were shown in orange. The 0% control group was stained green, whereas the area was predominantly colored red in the experimental groups. Both the red region and the number of dead bacteria grew as the number of reinforcing fillers rose, indicating that the antibacterial action grew.

#### 2.2.3. Bioactive Properties

The specimens were soaked in SBF for different durations. The surface morphology of the 0% control group showed that the fillers were evenly distributed in the resin matrix, but no mineralized nodules formed on the surface (Figure 5A,E). In the 2.5% group and 5% group, dispersed apatite particles could be observed (Figure 5B,C,F). With the time of soaking extended and the proportion of reinforcing fillers increased, the surface of the resin composite was covered by thick and dense layers (Figure 5D,G,H). The layers were formed by the tight accumulation of many globular mineralized nodules. 

Carbon (C), oxygen (O), silicon (Si), calcium (Ca), and phosphorus (P) were detected on the surface of the DRCs by EDS mapping. With the time and filler content increasing, Ca and P from the precipitated apatite layer showed significant intensities on the sample surface. Additionally, the Ca/P ratios of the surface layers were calculated and labeled in the figures. 

### 2.3. Cell Compatibility

By tracking the growth of L929 fibroblasts in the resin extracts, the cytotoxicity was assessed. In Table 3, Table 4 and Table 5, the relative growth rate (RGR) values and cytotoxicity levels at various time points are displayed. The RGR results in this investigation were all higher than 75%. The DRCs met the in vitro safety criteria, with the cytotoxicity level being 0 or I. 

## 3. Discussion

In clinical treatment, the accumulation of the oral biofilm and the degradation process of the resin led to secondary caries and caused the failure of the treatment [38,39,40]. To avoid the restorations failing too soon, the bacterial acid attack on the biomaterials’ surfaces at the bonded interfaces must be reduced [41]. It has been suggested that the most effective approach to deal with this problem is to add bioactive antimicrobial and remineralization agents [42]. The creation of bacterial acids and enzymes is inhibited by removing pathogenic bacteria, while the synthesis of new minerals may allow the gaps and faults around the resin to mend. The evolution of the functional biomaterial brings out an effective way to confer these properties to DRCs [43]. In recent years, TiO_2_-based materials have been widely studied as photocatalysts and antibacterial agents [44]. Currently, there are some investigations into the use of TiO_2_ in dental applications [45,46]. Herein, in this study, we aimed to synthesize a Sr-N-TiO_2_ composite via a sol-hydrothermal approach and mix it with HA as the fillers to modify the DRCs, eventually exploring the application potential of the multifunctional DRCs in the dentistry area.

### 3.1. Characterization of Sr-N-TiO_2_ Powders

Numerous polymorphs of TiO_2_ are found naturally in minerals, such as anatase (A), rutile (R), and brookite. Anatase is generally agreed to be the most active for photocatalysis based on experimental findings [47]. By using SEM, TEM, EDS, XPS, XRD, Raman, and UV–Vis, the phase composition and surface morphology of the produced composites were characterized. These outcomes demonstrated that the Sr-N-TiO_2_ nanoparticles were successfully synthesized and identified as anatase-phase TiO_2_. Because of the differing chemical states, the XPS analysis in Figure 1G indicated that the spectra of O 1 s of doped TiO_2_ corresponded to Ti-O-Ti and Ti-O-H peaks [48]. In general, dopant ions lead to local defects in the crystal, raise the energy barrier of mutual diffusion between the grains, restrict direct contact between the grains, and impede the expansion of the grains, resulting in smaller grain sizes and a higher diffraction half-width peak in XRD. The flaws of doped TiO_2_ were the active photocatalytic sites to improve the absorption of visible light. The result was also reconfirmed by UV–Vis. After doping, the observations in Figure 1J indicated a slight red shift and apparent visible-light absorption, which were attributable to the altered bandgap of TiO_2_. The bandgap energy was reduced as a result of N 2p and O 2p hybridization. Another factor was the partial replacement of Ti^4+^ by Sr^2+^ in the TiO_2_ crystal lattice, which increased the absorption strength by absorbing a longer wavelength of light. These characterizations of doped TiO_2_ were also consistent with the conclusions reached by Huang [49].

### 3.2. Physicochemical Properties of DRCs

The properties of polymers are achieved by the conversion of double bonds (C = C) to single bonds (C-C) in aliphatic chains during the free radical polymerization of resin-based materials [50]. Figure 2A reveals that the peaks gradually reduced with the curing time. The molecular structure of the monomer, inorganic filler, resin viscosity, light curing duration, and photoinitiating system, among other factors, all have a significant impact on the DC rate [51,52]. In this work, the DC rates of the experimental groups decreased as the proportion of the enhanced filler increased with a constant total filler mass. 

In accordance with the ISO-4049 standard, the CD of dental restoration materials should be more than 1.5 mm to meet the clinical treatment needs [53]. Additionally, the CD of the resin ranged from 4.36 ± 0.18 mm to 5.10 ± 0.19 mm, which satisfied the curing depth specified in the standard. The above results indicate that the fundamental physical properties of the novel composite resin meet the criteria for clinical applications. According to the above findings, the innovative composite resin’s fundamental physical characteristics are adequate for clinical use.

An essential physicochemical characteristic of biomaterials is surface wettability, which can affect protein adsorption and therefore affect cell attachment and behavior [54,55]. According to studies, cells may adhere and multiply at their greatest rate when grown on surfaces with a 70° contact angle [56]. This work measured the WCA values to evaluate hydrophobicity. The variation in WCA might be related to the filler. Different volumes for the same mass fraction came from the reinforced filler’s smaller diameter than the ordinary filler. The WCA of the resin over 70° had an inhibitory effect on bacterial adhesion. On the other hand, TiO_2_ and n-HA have a certain degree of hydrophilicity [57,58]. Thus, they could release ROS and ions into the liquid. With the increases in the mass fraction of TiO_2_ and n-HA, the influence of hydrophilicity gradually overtook the influence of diameter. This might be the reason that the WCA in the 7.5% group was lower than that in the 2.5% and 5% groups. Moreover, the lower WCA provided a new approach from another perspective to explain the highest antibacterial rates in the 7.5% group. The surface of the 7.5% group DRC has a larger contact area with the liquid; that is to say, the effective antibacterial ingredients are easier to release.

### 3.3. Antibacterial Properties and Antibacterial Mechanism of DRCs

The prevention of secondary caries is possible with effective antimicrobial activity, which is crucial for dental restoration. According to various epidemiological research, *S. mutans* is the main microbiological cause of dental caries [59], as well as the most common pathogen which was isolated from the dental plaque in humans [60]. Thus, *S. mutans* was selected as the test organism to examine the potential efficacy of the loaded DRCs. Moreover, crystal violet can be absorbed by living cells and stained, enabling a quantitative analysis to assess the growth and reproduction of live bacteria [61]. According to Figure 3A–C, a decrease in bacteria was seen as the TiO_2_ increased. These findings implied that the DRCs in this work could prevent the growth and colonization of cariogenic bacteria.

To further clarify the photocatalytic antibacterial mechanism, additional research was conducted. A 1,3-diphenylisobenzofuran (DPBF) probe is an effective experimental tool to confirm the release of ROS [62,63]. N-acetyl cysteine (NAC) is known as an antioxidant to reduce ROS [64]. The decreasing absorption peak of DPBF at 410 nm in Figure 3D confirmed the generation of ROS. As shown in Figure 3E,F, the positive group with Sr-N-TiO_2_ was no doubt the one with the lowest colony counting number (*p* < 0.0001). As for the experimental group that co-cultured with both Sr-N-TiO_2_ and NAC, the addition of NAC reduced part of the antibacterial properties of Sr-N-TiO_2_. So, the number was greater than that of the positive group, but lower than that of the other two groups.

Above results demonstrated that ROS was the main mechanism for the antibacterial activity. The doped TiO_2_ stimulates the bandgap, which causes the movement of free electrons and electron holes and results in the release of ROS. In addition, ROS attack the bacteria, leading to protein denaturation and the electron mediators’ release, which breach cell membranes, cause cytoplasmic leakage, and ultimately cause death. This also accords with the SEM results, which showed the damage to bacteria. Overall, the results demonstrated the antibacterial effect of TiO_2_, which was broadly consistent with that of other researchers [22,65]. Based on the above experiments, the experimental groups had a significant antibacterial effect, and the DRCs had an excellent bactericidal effect and antibacterial adhesion effect. Nonetheless, the duration of the antimicrobial properties requires further investigation.

### 3.4. Mineralization Properties of DRCs

SEM and EDS were performed to confirm the formation of the apatite precipitation. Figure 5 presented the different results of mineralization in each group. As the proportion of the reinforcing fillers increased, the number of mineralized nodules on the surface and the volume increased. It was also proportional to time. It appeared that the increase in n-HA in the DRCs was sufficient to induce more precipitation. 

Furthermore, EDS mapping detected C, O, Si, Ca, and P in the precipitate. The C and Si might be from the resin matrix and the SiO_2_ filler. A small amount of Ca and P in the 0% group might be due to the deposition of SBF, which was not thoroughly washed out. Moreover, an elemental analysis of the minerals forming on the DRCs revealed that the molar ratio of Ca/P ranged from 1.44 to 1.83, close to the ratio of 1.67 of hydroxyapatite [66]. The observations were reasonable to assume that the DRCs in this study could spontaneously form mineral precipitates approximated to HA with SBF immersion. The exchange of calcium and phosphorus ions at the boundary between the bioactive material and SBF solution caused the generation of the apatite layer, which defines the bioactivity of a material in vitro [67]. The apatite layer formed around the DRCs could narrow the gap between the resin composite and the tooth. In other words, it could further reduce the potential for bacterial accumulation and secondary caries. Notably, the precipitation and deposition of hydroxyapatite into dental structures were not covered in this study. Further studies are required to explore the mineralization ability of modified DRCs on the dentin surface.

### 3.5. Cell Compatibility of DRCs

Due to the unique wet environment of the oral cavity, the nonpolymerized components of dental materials can be released into the mouth [68]. Therefore, the dental restoration should have good cell compatibility besides antibacterial and mineralization activities. The cytotoxicity test showed a high biosafety level of the DRCs in vitro, which provided the possibility of clinical use. On the other hand, it still remained to be seen whether the released ROS would disrupt the normal bacteria flora balance [69].

## 4. Materials and Methods

### 4.1. Synthesis and Characterization of the Modified Nanoparticles

Sol-hydrothermal method was used to create Sr-N co-doped TiO_2_ nanoparticles [70]. In order to carry out the hydrolysis, a mixed solution containing 5 mL of tetrabutyl titanate (Tianjin Guangfu Fine Chemical Research Institute, Tianjin, China) and 5 mL of anhydrous ethanol was first added dropwise to another hybrid solution made up of 20 mL of anhydrous ethanol, 5 mL of water, 1 mL of 70% nitric acid, 17.16 mL of ammonia solution, and 0.3109 g of strontium nitrate. The other products were purchased from Beijing Chemical Company, Beijing, China. By stirring constantly for 2 h, the yellowish translucent sol was created. In a stainless-steel jar, the as-prepared sol was maintained at 160 °C for 6 h. Then the production was immersed in anhydrous ethanol and sonicated for 30 min, centrifuged at high speed for 10 min. Finally, the production was dried at 60 °C for 24 h, and obtained by calcining at 450 °C for 2 h to produce Sr-N-TiO_2_ nanoparticles. All reagents were analytical grade and used without further purification.

The field emission scanning electron microscope (FE-SEM, JEOL JSM-6700F, Tokyo, Japan) operated at 3.0 kV was used to measure the surface morphology of the sample. We redispersed Sr-N-TiO_2_ NPs in ethanol, dropped them on carbon-coated copper grids, and used a transmission electron microscope (TEM, JEOL, Tokyo, Japan) to observe them. X-ray photoelectron spectroscopy (XPS, Thermo Kalpha, Waltham, MA, USA) was used to determine the valence states and chemical makeup of elements. Through the use of X-ray diffraction (XRD, X’Pert PRO MPD, Almelo, The Netherlands), the crystal structure of the modified TiO_2_ sample was identified. A LabRAM ARAMIS system was used to obtain the Raman spectra. The He-Ne laser light at 633 nm was used as the excitation source. Ten-second accumulations led to data gathering. On a Shimadzu UV-3600 UV–Vis (Kyoto, Japan) spectrophotometer, the electronic absorption spectra were captured.

### 4.2. Preparation of DRCs

The resin matrix was synthesized, containing 49 wt% bisphenol-A-glycidyldimethacrylate (Bis-GMA) and 49 wt% triethylene glycol dimethacrylate (TEGDMA), with 1 wt% of camphorquinone (CQ) and 1 wt% of 2-(Dimethylamino)ethyl methacrylate (DMAEMA). All products were analytical grade without further purification from Sigma Aldrich Chemical Co., St. Louis, MO, USA. The Sr-N-TiO_2_ nanoparticles were mixed up with nano-hydroxyapatite (n-HA, 20 nm, Emperor Nano, Nanjing, China) as reinforcing fillers, and silicon dioxide (SiO_2_, 5 μm, Aladdin, Shanghai, China) was the traditional filler. Then they were mixed into the resin matrix with the SpeedMixer DAC 150 (FlackTek Inc., Shanghai, China). The compositions of the fillers in each group are presented in Table 1, and the load was maintained at 60 wt%.

### 4.3. Physicochemical Properties

#### 4.3.1. Degree of Conversion

With the help of a Fourier-transform infrared spectrometer (FT-IR, Bruker VERTEX 80v, Salbruken, Germany), the degree of conversion (DC) was ascertained. Firstly, we applied a small amount of the DRCs to the KBR slice and analyzed it. Following a 20 s light-curing unit exposure, it was examined. The process was repeated three times to measure separately the spectra at 20 s, 40 s, and 60 s.

Two absorption bands were examined in order to determine the DC. After exposure to radiation, the acrylate double bonds’ (C=C) absorbance intensity at 1636 cm^−1^ decreased. The internal standard was the carbonyl (C=O) absorbance peak at 1720 cm^−1^. Equation below was used to compute the DC:$$\mathrm{DC}=\left[1-\frac{{\left(\mathrm{AC}=C/\mathrm{AC}=O\right)}_{t}}{{\left(\mathrm{AC}=C/\mathrm{AC}=O\right)}_{0}}\right]\times 100\%$$
where AC=C and AC=O represented, respectively, the absorbance peaks at 1636 cm^−1^ of C=C and 1720 cm^−1^ of C=O. The absorbances of the functional group before and after light-curing were, respectively, (AC=C/AC=O)_0_ and (AC=C/AC=O)_t_. Measurements were made in the dark and repeated three times.

#### 4.3.2. Curing Depth 

The method described in ISO 4049: 2009 was used to measure the curing depth (CD) of composites. The cylindrical specimens of each composite (d = 4.0 mm, h = 10.0 mm) were made. The light-curing equipment photopolymerized the composite for 20 s, starting from one side. The uncured material was removed after irradiation. A digital micrometer was used to measure the height of the cured composite. Each measurement was carried out five times.

#### 4.3.3. Water Contact Angle

Sessile drop analysis was used to calculate the water contact angle (WCA) using the OCA 15EC goniometer from Data Physics in Filderstadt, Germany. We used fine sandpaper to smooth out the samples. The surface of the sample was then sprayed with 5 μL of deionized water. The process was repeated three times.

### 4.4. Antibacterial Properties and Antibacterial Mechanism

#### 4.4.1. Bacterial Culture

In a bacteria incubator (SLI-1200, SANYO, Tokyo, Japan), the frozen precultured *Streptococcus mutans* (*S. mutans*, UA159) were revived and injected on an agar plate with sterilized brain–heart infusion broth (BHI, Oxoid, Basingstoke, UK) culture media. Before the experiment, a single bacteria colony was added to a new sterilized BHI and cultured for 24 h. By altering the absorbance at 600 nm of the bacteria strain with a microplate reader (Synergy HT, Biotek, Winooski, VT, USA), the concentration of the bacteria was managed.

#### 4.4.2. Colony-Forming Units (CFU) Counting

The 6 mm × 2 mm cylindrical specimens were co-cultured with 2 mL bacterial suspension (1 × 10^7^ CFU/L) for 24 h. To get rid of non-adherent bacteria, PBS buffer was used to rinse the co-cultured specimens gently. The adherent *S. mutans* were collected, diluted, and inoculated on BHI broth agar. The number of colonies that developed on each plate was then counted after a 24 h anaerobic incubation period at 37 °C. 

The following equation was used to calculate the antibacterial rate (AR) for each group: AR = (CFU_0_ − CFU)/CFU_0_ × 100% (where CFU_0_ represents the average colony count of the control group and CFU represents the average colony count of the experimental group). The above experiment recurred three times.

#### 4.4.3. Crystal Violet Staining Assay

The 24 h co-cultured specimens were dipped for 30 min into a glutaraldehyde solution (Solarbio, Beijing, China) and stained for 20 min by crystal violet solution. Then 95% alcohol was used to decolorize the samples and they were transferred to a new 96-well plate to record the OD values at 600 nm.

#### 4.4.4. Detection of ROS Release

The release of ROS under different processing was analyzed with a DPBF probe (Macklin, Shanghai, China). The Sr-N-TiO_2_ nanoparticles were prepared in PBS solution with a concentration of 2.0 mg/mL. The solution was transferred to 96-well plates, and placed under LED light for 0, 1, 3, and 5 min, respectively. The reduction in the characteristic peak of DPBF at 410 nm by UV–Vis absorption spectroscopy demonstrated the generation of ROS.

#### 4.4.5. CFU Counting with NAC

The growth of colonies with the addition of NAC (Aladdin Industrial Corporation, Shanghai, China), which was the ROS scavenger, was examined by the CFU method. The Sr-N-TiO_2_ nanoparticles were prepared into BHI solution with a concentration of 4 mg/mL. The groups in the 96-well plate were as follows: the blank control group (50 μL BHI), the negative control group (25 μL NAC and 25 μL BHI), the positive control group (25 μL Sr-N-TiO_2_ and 25 μL BHI), and the experimental group (25 μL NAC and 25 μL Sr-N-TiO_2_). Then 50 μL bacterial suspension (1 × 10^7^ CFU/L) was added per well. According to GB/T 21510-2008, the plate was incubated at 37 °C with constant shaking for 4 h and co-cultured for 24 h. The *S. mutans* were diluted and inoculated on BHI broth agar. The number of colonies was counted and analyzed.

#### 4.4.6. SEM of the Bacteria Attached to the Surface

The preparation conditions were the same as 4.4.2. After removing the non-adhered bacteria, the bacteria were fixed for 12 h with 2.5% glutaraldehyde and then dehydrated for 15 min at each concentration using a gradient ethanol series of 30%, 50%, 70%, 80%, 90%, and 100% (*v*/*v*). Before SEM analysis, the samples were dried and gold-sprayed. 

#### 4.4.7. Bacterial Live/Dead Staining

The stain was mixed with SYTO 9 dye and propidium iodide (PI) dye (Bestbio, Shanghai, China) in a 1:1 ratio. The specimens were treated in the same way as 4.4.2, then examed using confocal laser scanning microscopy (CLSM) after staining for 30 min.

### 4.5. Remineralization Properties

The 6 mm × 2 mm cylindrical specimens were incubated at 37 °C with constant shaking in 5 mL synthetic saliva (SBF, Solarbio, Beijing, China). At predetermined intervals (14 d and 28 d), samples were taken out and dried at 50 °C. By using a FE-SEM with EDS, the morphological surface alterations and element quantification were carried out.

### 4.6. Cytotoxic Properties

Following the instructions of ISO 10993-5 standard, the samples were soaked in high-glucose medium (DMEM, HyClone, Logan, UT, USA) to set up the extraction solution for 24 h at 37 °C. The following five groups were marked: blank control group, 0% group, 2.5% group, 5% group, and 7.5% group. L929 cells (Cells Resource Center, Shanghai Institutes of Biological Science, Shanghai, China) in the logarithmic growth phase were placed in the 96-well plates with 2000 cells per well and incubated for 24 h in the cell incubator. After removing the supernatant, 200 μL extraction solution per well was added. On day 1, day 2, and day 3, cell counting kit-8 (CCK-8, Beyotime, Shanghai, China) solution was added (200 μL per well) and incubated for 1 h, 2 h, and 3 h. Later, to assess cell proliferation, the OD values were determined with a microreader (Bio-Rad 680) at a 450 nm wavelength.

The equation can be used to calculate: RGR = OD_0_/OD_t_ × 100% (where OD_0_ is the average OD value of the control group and OD_t_ is the average OD value of the experimental group). Based on ISO standards and the US Pharmacopoeial Convention [71] in Table 6, the RGR and toxicity grades were assessed.

### 4.7. Statistical Analysis

The software GraphPad Prism 8 (GraphPad, San Diego, CA, USA) was used to obtain the statistical analysis. All data were recorded as mean ± standard deviation (Mean ± SD). A one-way analysis of variance was used to perform multiple group comparisons. * *p* < 0.05, ** *p* < 0.01, *** *p* < 0.001, and **** *p* < 0.0001. Ns indicated not significant.

## 5. Conclusions

In this study, we synthesized anatase-phase Sr-N-TiO_2_ to increase its photocatalytic activity, then mixed it up with n-HA fillers as the reinforcing fillers to prepare a novel multifunctional DRC. The DRC complied with clinical standards and showed high biosafety, with the cytotoxicity level being 0 or I. In recent years, previous studies have largely focused on antibacterial or remineralized DRCs. In contrast, our work effectively identifies both the antibacterial and mineralization properties of DRCs by the synergistic effect of Sr-N-TiO_2_ and n-HA. The study indicated a 98.96% antibacterial rate and hydroxyapatite-like remineralization, which was helpful to lengthen the longevity of the DRC in the clinic. Consequently, we can conclude that the DRC with Sr-N-TiO_2_ and n-HA fillers is expected to be an ideal filling material for caries restoration, and it has broad prospects for clinical applications.

## Figures and Tables

**Figure 1 ijms-24-01274-f001:**
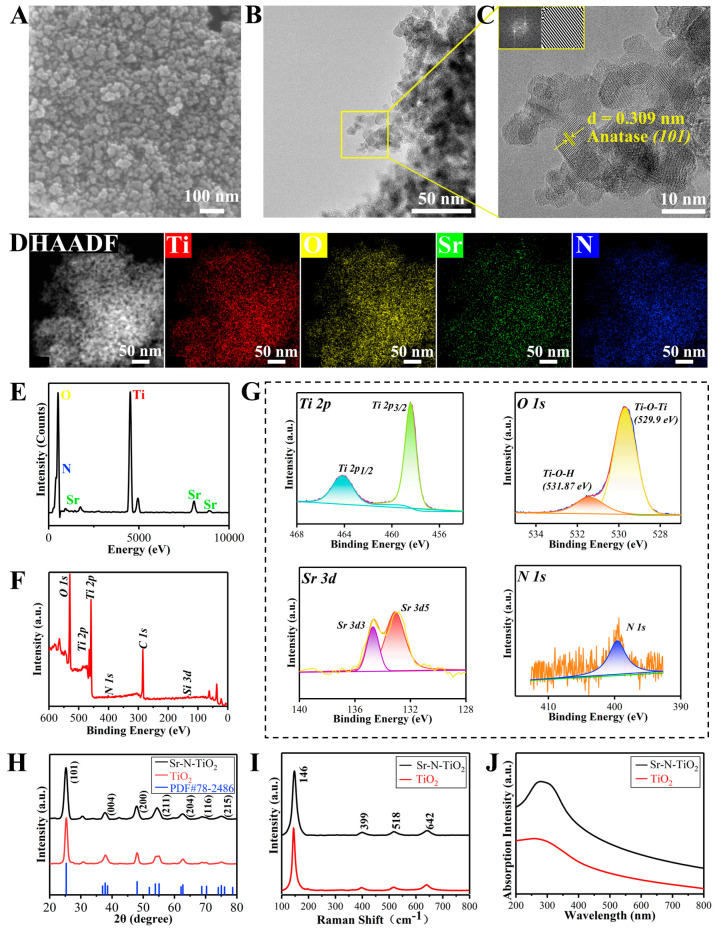
Characterization of Sr-N-TiO_2_ nanoparticles. (**A**) Field emission scanning electron microscope (FE-SEM) micrographs; (**B**,**C**) transmission electron microscopy (TEM) micrograph of the Sr-N-TiO_2_ powders. (**D**) High-angle annular dark field (HAADF) image and (**E**) Energy-dispersive X-ray spectroscopy (EDS) spectrum of Ti, O, Sr, and N elements in Sr-N-TiO_2_. (**F**,**G**) X-ray photoelectron spectroscopy (XPS) pattern. (**H**) X-ray diffraction (XRD) pattern, (**I**) Raman spectra, and (**J**) UV–Vis absorption spectra of synthesized Sr-N-TiO_2_ and TiO_2_.

**Figure 2 ijms-24-01274-f002:**
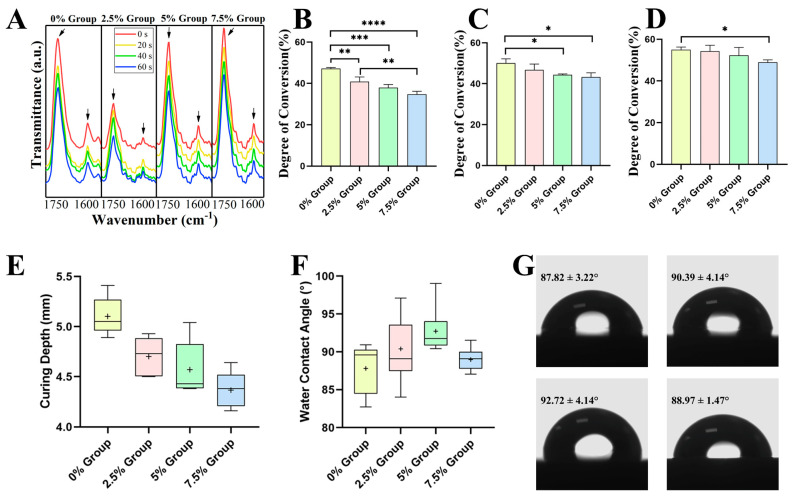
Physicochemical properties of DRs. (**A**) FT-IR spectra between 1750 cm^−1^ and 1600 cm^−1^ (the arrows show the absorbance peaks at 1636 cm^−1^ of C=C and 1720 cm^−1^ of C=O) and the DC rate with curing for (**B**) 20 s, (**C**) 40 s, (**D**) 60 s in each group. (**E**) CD at curing for 20 s. (**F**,**G**) Water contact angle (WCA) of 5 μL deionized water on the DRC surface. * *p* < 0.05, ** *p* < 0.01, *** *p* < 0.001, and **** *p* < 0.0001.

**Figure 3 ijms-24-01274-f003:**
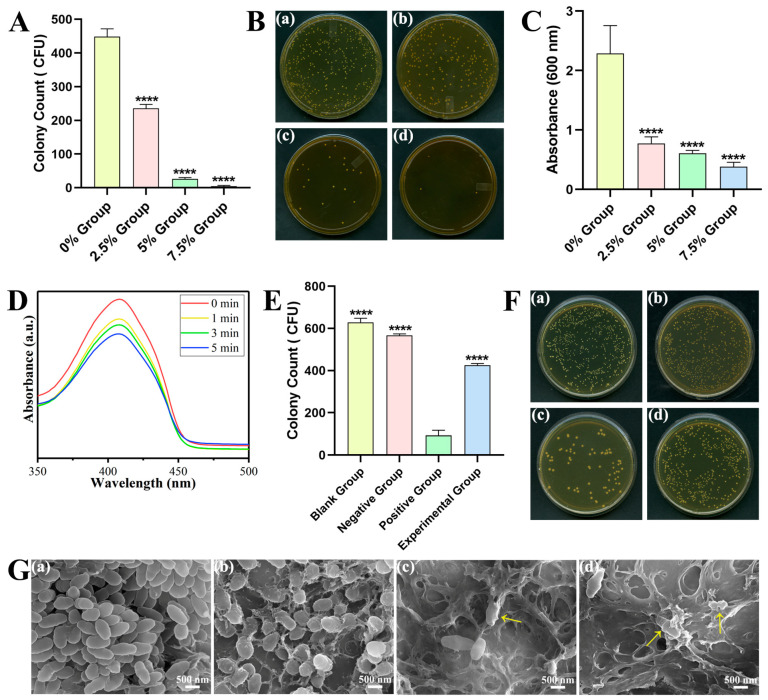
Antibacterial properties and antibacterial mechanism of DRCs. (**A**,**B**) Colony-forming units (CFU) counting. a: 0% Group, b: 2.5% Group, c: 5% Group, d: 7.5% Group. (**C**) Crystal violet staining assay. (**D**) The characteristic peak of DPBF at 410 nm under LED light for 0, 1, 3, and 5 min. (**E**,**F**) CFU counting with NAC. Blank group (BHI), the negative group (NAC), the positive group (Sr-N-TiO_2_), and the experimental group (NAC and Sr-N-TiO_2_). a: 0% Group, b: 2.5% Group, c: 5% Group, d: 7.5% Group. (**G**) SEM image of the bacteria on the surface; a: 0% Group, b: 2.5% Group, c: 5% Group, d: 7.5% Group. The yellow arrows show the unusual bacteria. **** *p* < 0.0001.

**Figure 4 ijms-24-01274-f004:**
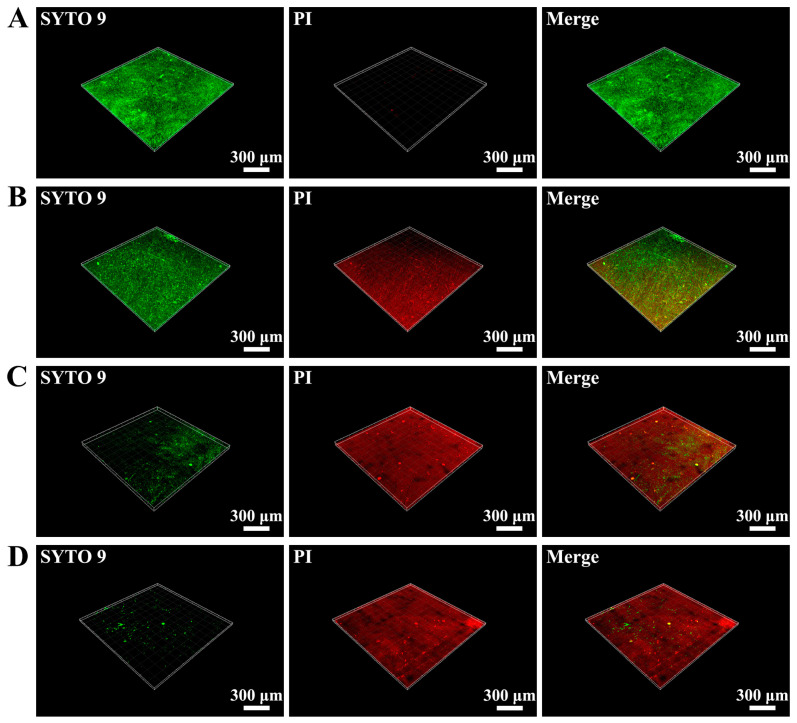
Live/dead staining of bacterial biofilm on DRCs’ surface. (**A**) 0% group, (**B**) 2.5% group, (**C**) 5% group, (**D**) 7.5% group. The damaged bacteria were stained red by PI dye, and the living bacteria were stained green by SYTO 9 dye.

**Figure 5 ijms-24-01274-f005:**
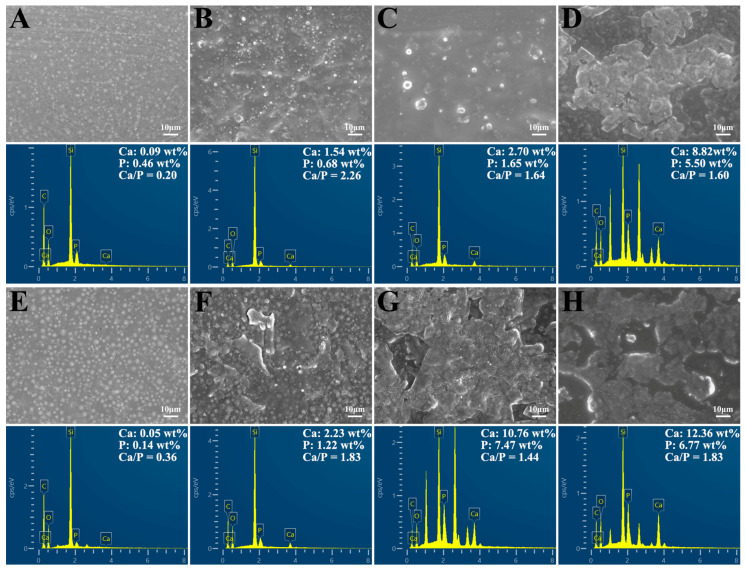
The morphological surface alterations and element quantification of (**A**–**D**) 14 d and (**E**–**H**) 28 d using an FE-SEM with EDS. (**A**,**E**) 0% group, (**B**,**F**) 2.5% group, (**C**,**G**) 5% group, (**D**,**H**) 7.5% group. The Ca/P ratios are labeled in the EDS figures.

**Table 1 ijms-24-01274-t001:** Formulations of fillers in DRC of each group. The total filler load amounted to 60 wt%.

Group		Reinforcing Fillers (wt%)	Traditional Fillers (wt%)
Control Group	0% Group	0	60
Experimental Groups	2.5% Group	2.5	57.5
	5% Group	5	55
	7.5% Group	7.5	52.5

**Table 2 ijms-24-01274-t002:** The degree of conversion (DC) and curing depth (CD) of each group.

Groups	DC (%)			CD (mm)
	20 s	40 s	60 s	
0% Group	47.12 ± 0.56	50.13 ± 2.06	55.04 ± 1.21	5.10 ± 0.19
2.5% Group	40.84 ± 2.30	46.71 ± 2.93	54.30 ± 2.83	4.70 ± 0.19
5% Group	37.99 ± 1.50	44.39 ± 0.37	52.27 ± 3.80	4.57 ± 0.28
7.5% Group	34.75 ± 1.44	43.21 ± 2.13	48.98 ± 1.13	4.36 ± 0.18

**Table 3 ijms-24-01274-t003:** RGR and cytotoxicity grade at 1 d.

Group	RGR (%)			Cytotoxicity Grades		
	1 h	2 h	3 h	1 h	2 h	3 h
0% Group	85.88	93.25	89.22	I	I	I
2.5% Group	90.28	95.75	92.24	I	I	I
5% Group	78.52	87.31	86.28	I	I	I
7.5% Group	76.25	85.45	87.48	I	I	I

**Table 4 ijms-24-01274-t004:** RGR and cytotoxicity grade at 2 d.

Group	RGR (%)			Cytotoxicity Grades		
	1 h	2 h	3 h	1 h	2 h	3 h
0% Group	109.14	109.17	111.82	0	0	0
2.5% Group	84.88	90.91	91.24	I	I	I
5% Group	81.08	100.69	98.63	I	0	I
7.5% Group	79.39	94.95	91.73	I	I	I

**Table 5 ijms-24-01274-t005:** RGR and cytotoxicity grade at 3 d.

Group	RGR (%)			Cytotoxicity Grades		
	1 h	2 h	3 h	1 h	2 h	3 h
0% Group	142.31	123.28	117.04	0	0	0
2.5% Group	135.44	106.72	101.51	0	0	0
5% Group	139.10	128.92	117.79	0	0	0
7.5% Group	120.83	106.61	101.05	0	0	0

**Table 6 ijms-24-01274-t006:** Classification of cytotoxicity.

RGR (%)	Toxicity Grade	Safety Standards
≥100	0	Safe
75–99	I	Safe
50–74	II	Insecurity
25–49	III	Insecurity
1–24	IV	Insecurity
<1	V	Insecurity

## Data Availability

Data supporting reported results are available from the authors.

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
