# Peer review of "A Multifunctional Dental Resin Composite with Sr-N-Doped TiO2 and n-HA Fillers for Antibacterial and Mineralization Effects"

_ijms, 2023, doi:10.3390/ijms24021274_

Round 1

Reviewer 1 Report

Good work. Please follow the reviewer suggestion.

Author Response

Dear Editors and Reviewers:

Thank you for your letter and for the reviewers’ comments concerning our manuscript entitled " A Multifunctional Dental Resin Composite with Sr-N-Doped TiO2 and n-HA Fillers for Antibacterial and Mineralization Effects" (ijms-2088306). Those comments are all valuable and very helpful for revising and improving our paper, as well as the important guiding significance to our researches. We have studied comments carefully and have made correction which we hope meet with approval. Revised portion are marked in red in the paper. The main corrections in the paper and the responses to the reviewer's comments are marked in bold as following:

Responses to the reviewer's comments:

Reviewer #1:

  1. Submitted article is within the scope of the journal:

Response: We appreciate your warm work earnestly and hope that the corrections will meet with approval.

  1. Abstract of the article is suitable. More theoretical information is provided whereas quantitative values are more required. Revised the abstract properly if more quantities include, it will be better.

Response: We have re-written the abstract according to your useful suggestion. More quantitative values were included in this part.

  1. Introduction is written simply; most recent research and innovation on biocompatibility should be reviewed to show the gap in knowledge. The introduction should be rewritten to show the highlights and novelty of the work.

Response: We thank the reviewer for the valuable comments. We have resived the text to address your concerns and hope that it is now clearer. Please see Page 2, Line 66-75 and Line 81–86, and we have cited more recent references as [26-37].

  1. What edge does this paper have over the existing works? This needs to be presented as a novelty statement in the introduction section.

Response: Thanks for your suggestions. We are sorry that the highlights and the novelty statement were indistinct due to our unclear writing. We have re-written this part in Page 2, Line 76-77, Line 87, and Line 97-99. In conclution, firstly, we applied a better antibacterial agent, TiO2 which was codoped with other elements. Then, in contrast to the previous studies which mostly focused on the single function, we combined antimicrobial agents with remineralization agents together to provide the resin with more multiple functions.

  1. In Introduction section, Sentence no. 62, it was mentioned that TiO2 can be activated by absorbing the Ultra high band gap energy radiation……………………. what other sources may be referenced to claim this sentence, may be added as reference here.

Response: We sincerely appreciate the valuable comments. We have checked the literature carefully and cited Guo’s work, referenced as [21] in Page 2, Line 59.

  1. In this manuscript, methodology section is kept in section 4. Please add methodology section after introduction section.

Response: We appreciate your suggestions and it may be clear to change the sequence. However, the template file of the journal keeps the Materials and Methods in section 4. We are not sure if the change meets the requirements or not.

  1. The table 1 need to plot again.

Response: Thank you for the comments. We have re-plot the table 1 according to your suggestion, and hope that it is clearer now. Please tell us if we could revise the table more.

  1. The figure 2 G shows contact angle. please mention the reason behind decreasing value at 7.5% group. The figure 2G need to again plot, mentioning, what’s the standard liquid has been used to conduct the test.

Response: According to your opinion, we have reviewed lots of articles again and stated the resons in Page 11, Line 304-311. Besides, we used 5 mL deionized water as the standard liquid to conduct the WCA test and have marked it in figure caption in Page 6, Line 176.

  1. It is suggested to keep section 4 (materials and methods) just after introduction section.

Response: We appreciate your suggestions and it may be clear to change the sequence. However, the template file of the journal keeps the Materials and Methods in section 4. We are not sure if the change meets the requirements or not.

  1. In section 3 authors have well written the discussion of the research work in and explained the results properly. But it is suggested to have mechanical strength of the composite and also the mechanical strength of the dental parts may be matched in order to prevent stress shielding effect.

Response: We sincerely appreciate the valuable comments. This is really a question to think about. We agree with the reviewer that it would be helpful to have some mechanical strength tests. However, the dental resin composite composed of Bis-GMA, TEGDMA, and SiO2 was a very traditional and well-proven solution in decades years. In this study, we prepared DRC with Bis-GMA and TEGDMA in a 1:1 ratio, and the total filler load amounted to 60 wt%. The fillers were composed with SiO2 (5 mm) and nano-sized particles. We reviewed some articles and make a brief summary here. In ISO 4049–2009 (International Standards Organization, Geneva, Switzerland), the minimum value was 80 MPa to meet clinical requirements. Wang et al. [1] prepared DRC with Bis-GMA and TEGDMA (50 wt%, 50 wt%), and SiO2 (1-8 mm) as fillers (60 wt%), which was the same ratio as this work. The results they came up with showed that the flexural strength (FS) was 86.52 ± 4.88 Mpa. And FS of the groups composed with SiO2 nanoparticles and SiO2 microparticles increased. They proposed that the combination of nanofillers and microfillers could improve the mechanical properties. Meanwhile, Liu et al. [2] also prepared DRC in the same solution. The FS of their work was also more than 80 Mpa. It is worth noting that they mixed HA in DRC and confirmed that HA enhanced the mechanical properties of DRC due to its similar hardness to that of teeth. Therefore, We consider that these findings could support that the mechanical properties of DRC in our work might be superior than 80 Mpa.

  1. A conclusion section may also include the positive results suggested be added in revised article.

Response: Thanks for your suggestion. It would be more complete to add some positive results in the conclusion part. We have revised this part in Line 512-516.

  1. Highlight the future scope of present research work.

Response: We appreciate your suggestions and revise the future scope in conclusion in Line 512-516.

  1. In discussion section, it is difficult to ascertain the meaning or significance of the authors’ findings due to unclear writing. A more simplified narrative is required.

Response: Thank you for your useful comments. We have revised part of the discussion section and correlated the discussions with the results, particularly the “3.3 Antibacterial Properties and Antibacterial Mechanism of DRCs” section in Line 319-321 and 325-330.

  1. Pay close attention to style guidelines (formatting for references/citations in text and equation formatting).

Response: We appreciate your careful work. We have carefully checked the manuscript and corrected the formatting for ciations at the end of the article.

We tried our best to improve the manu and made some changes in the manu. These changes will not influence the content and framework of the paper. And here we did not list the changes but marked in red in revised paper.

We appreciate for your warm work earnestly, and hope that the correction will meet with approval.

Once again, thank you very much for your couments and suggestions.

  1. Wang, X.; Cai, Q.; Zhang, X.; Wei, Y.; Xu, M.; Yang, X.; Ma, Q.; Cheng, Y.; Deng, X., Improved performance of Bis-GMA/TEGDMA dental composites by net-like structures formed from SiO2 nanofiber fillers. Mater Sci Eng C Mater Biol Appl 2016, 59, 464-470.
  2. Liu, F.; Jiang, X.; Zhang, Q.; Zhu, M., Strong and bioactive dental resin composite containing poly(Bis-GMA) grafted hydroxyapatite whiskers and silica nanoparticles. Composites Science and Technology 2014, 101, 86-93.

Reviewer 2 Report

The reported work entitled "A Multifunctional Dental Resin Composite with Sr-N-Doped  TiO2 and n-HA Fillers for Antibacterial and Mineralization Effects” is interesting. However, the manuscript can be accepted in Molecules after taking my concerns into account, as follows.
1. Many grammatical and typographical errors must be carefully corrected.

2. Abstract is not clear and very confusing to reader. Kindly rewrite the abstract.

3. Introduction need more reference from recent study.

4. What is new in your study.
5. Kindly correlated your discussion with your results.

6.  Write about the materials you have used in your research i.e. source, percentage purity, grade (laboratory or analytical grade)
7. Have you carried out the MIC for your research product??

Author Response

Dear Editors and Reviewers:

Thank you for your letter and for the reviewers’ comments concerning our manuscript entitled " A Multifunctional Dental Resin Composite with Sr-N-Doped TiO2 and n-HA Fillers for Antibacterial and Mineralization Effects" (ijms-2088306). Those comments are all valuable and very helpful for revising and improving our paper, as well as the important guiding significance to our researches. We have studied comments carefully and have made correction which we hope meet with approval. Revised portion are marked in red in the paper. The main corrections in the paper and the responses to the reviewer's comments are marked in bold as following:

Responses to the reviewer's comments:

Reviewer #2:

  1. Many grammatical and typographical errors must be carefully corrected.

Response: We appreciate your warm work. We have carefully checked the manuscript and corrected the grammatical and typographical errors.

  1. Abstract is not clear and very confusing to reader. Kindly rewrite the abstract.

Response: Thanks for your comments. We have rewritten the abstract according to your useful suggestion. And more results were included to make abstract more completely in Line 19-25.

  1. Introduction need more reference from recent study.

Response: We thank the reviewer for the valuable comments. We have resived the text to address your concerns and hope that it is now clearer. Please see Page 2, Line 66-75 and Line 81–86, and we have cited more recent references as [26-37].

  1. What is new in your study.

Response: Thanks for your suggestions. We are sorry that the highlights and the novelty statement were indistinct due to our unclear writing. We have re-written this part in Page 2, Line 76-77, Line 87, and Line 97-99. In conclution, firstly, we applied a better antibacterial agent, TiO2, which was codoped with other elements. Then, in contrast to the previous studies which mostly focused on the single function, we combined antimicrobial agents with remineralization agents together to provide the resin with more multiple functions.

  1. Kindly correlated your discussion with your results.

Response: Thank you for your useful comments. We have revised part of the discussion section and correlated the discussions with the results, particularly the WCA part in Line 304-311, and the “3.3 Antibacterial Properties and Antibacterial Mechanism of DRCs” section in Line 319-321 and Line 325-330.

  1. Write about the materials you have used in your research i.e. source, percentage purity, grade (laboratory or analytical grade)

Response: We appreciate your careful work. We have carefully checked the manuscript and supplied the materials informations including source and grade in “Materials and Methods” in section 4.

  1. Have you carried out the MIC for your research product??

Response: Thank you for pointing this out. We have not carried out the MIC tests. We agree that MIC would be useful to detect the antibacterial activity. Due to the limitations of COVID-19, we are sorry that we are not able to carry out the work. However, we believe the present results can still support the conclusion of this paper. And we also noted that there have been some publications proposed the MIC data of TiO2. We reviewed some articles and make a brief summary here. Besides, we are glad to supply MIC experiment if there is a chance.

Author ï¼ˆYear) Bacterial Species MIC References

J. Rajkumari et al. (2019)

Pseudomonas aeruginosa (P. aeruginosa)

31.25 μg/ml

[1]
Nasser et al.(2020)

Serratia marcescens (S. marcescens)

8 µg/mL

[2]
Nasser et al.(2020)

Escherichia coli (E. coli),

Pseudomonas aeruginosa (P. aeruginosa),

methicillin-resistant Staphylococcus aureus (MRSA)

32 µg/mL

[2]
Nasser et al.(2020)

Listeria monocytogenes (L. monocytogenes)

64 µg/mL

[2]
Nasser et al.(2020)

Candida albicans (C. albicans)

64 µg/mL

[2]

Kermani et al.(2020)

Candida species

128-256 µg/mL

[3]

We tried our best to improve the manuscript and made some changes in the manuscript. These changes will not influence the content and framework of the paper. And here we did not list the changes but marked in red in revised paper.

We appreciate for your warm work earnestly, and hope that the correction will meet with approval.

Once again, thank you very much for your couments and suggestions.

[1] Rajkumari, J.; Magdalane, C. M.; Siddhardha, B.; Madhavan, J.; Ramalingam, G.; Al-Dhabi, N. A.; Arasu, M. V.; Ghilan, A. K. M.; Duraipandiayan, V.; Kaviyarasu, K., Synthesis of titanium oxide nanoparticles using Aloe barbadensis mill and evaluation of its antibiofilm potential against Pseudomonas aeruginosa PAO1. J Photochem Photobiol B 2019, 201, 111667.

[2] Al-Shabib, N. A.; Husain, F. M.; Qais, F. A.; Ahmad, N.; Khan, A.; Alyousef, A. A.; Arshad, M.; Noor, S.; Khan, J. M.; Alam, P.; Albalawi, T. H.; Shahzad, S. A., Phyto-Mediated Synthesis of Porous Titanium Dioxide Nanoparticles From Withania somnifera Root Extract: Broad-Spectrum Attenuation of Biofilm and Cytotoxic Properties Against HepG2 Cell Lines. Front Microbiol 2020, 11, 1680.

[3] Ahmadpour Kermani, S.; Salari, S.; Ghasemi Nejad Almani, P., Comparison of antifungal and cytotoxicity activities of titanium dioxide and zinc oxide nanoparticles with amphotericin B against different Candida species: In vitro evaluation. J Clin Lab Anal 2021, 35, (1), e23577.

Round 2

Reviewer 2 Report

Comments and Suggestions for Authors

I would like to thank the authors for adequately addressing all the comments. 

I believe that the manuscript at its current form is in good shape for publication and that the quality of presentation, as well as the scientific soundness are significantly improved. I would suggest that the authors do a final revision and correct any minor spelling errors (subscripts/superscripts were necessary, space before units, etc.) before the final submission.     

Author Response

Dear Editors and Reviewers:

Thank you for your letter and for the reviewers’ comments concerning our manuscript entitled " A Multifunctional Dental Resin Composite with Sr-N-Doped TiO2 and n-HA Fillers for Antibacterial and Mineralization Effects" (ijms-2088306). Those comments are all valuable and very helpful for revising and improving our paper, as well as the important guiding significance to our researches. We have studied comments carefully and have made correction which we hope meet with approval. Revised portion are marked in red in the paper. The main corrections in the paper and the responses to the reviewer's comments are marked in bold as following:

Responses to the reviewer's comments:

1. I would suggest that the authors do a final revision and correct any minor spelling errors (subscripts/superscripts were necessary, space before units, etc.) before the final submission.

Respond: We appreciate your warm work. We have carefully checked the manuscript and corrected the grammatical and typographical errors, including the subscripts/superscripts and the space before units.

We tried our best to improve the manu and made some changes in the manu. These changes will not influence the content and framework of the paper. And here we did not list the changes but marked in red in revised paper.

We appreciate for your warm work earnestly, and hope that the correction will meet with approval.

Once again, thank you very much for your comments and suggestions.
